# Flight Phenology of *Spodoptera eridania* (Stoll, 1781) (Lepidoptera: Noctuidae) in Its Native Range: A Baseline for Managing an Emerging Invasive Pest

**DOI:** 10.3390/insects16080779

**Published:** 2025-07-29

**Authors:** Claudia Alzate, Eduardo Soares Calixto, Silvana V. Paula-Moraes

**Affiliations:** 1Entomology and Nematology Department, University of Florida, Gainesville, FL 32611, USA; calzate@ufl.edu; 2West Florida Research and Education Center, Department of Entomology and Nematology, University of Florida, Jay, FL 32565, USA; calixtos.edu@gmail.com; 3Department of Entomology, University of Nebraska-Lincoln, Lincoln, NE 68583, USA

**Keywords:** invasive species, population dynamics, phenological models, Florida panhandle, agricultural pest, pest management

## Abstract

The southern armyworm (*Spodoptera eridania*) is a moth species that has become a growing problem for farmers because it can feed on many different crops and is spreading to new parts of the world. It has also shown resistance to certain insect control methods, making it harder to manage. In this study, we monitored southern armyworm moths in the Florida Panhandle over six years to better understand when they are most active and what environmental conditions affect their numbers. We used traps that release a scent to attract moths and recorded how many were caught throughout the year. We found two clear periods when moths were most active: one in early spring and another larger one in the fall. These periods were linked to moderate temperatures, while rainfall and wind did not seem to affect moth numbers. These results are important because they help us predict when moth populations are likely to increase, which can allow farmers and pest control professionals to take action at the right time. By improving our ability to forecast pest outbreaks, this research supports more effective and responsible pest management strategies, which are essential for crop protection.

## 1. Introduction

Understanding the flight phenology and behavior of economic pests provides foundational information that could support future monitoring tools for IPM and IRM programs at the regional and continental scale, particularly considering pests that are highly polyphagous, migratory, and invasive and that have been associated with resistance to insecticides [1,2]. This is the case of *Spodoptera eridania* (Stoll, 1781) (Lepidoptera: Noctuidae), the southern armyworm. This species is considered one of the most polyphagous species within the genus *Spodoptera*, feeding on vegetative and reproductive stages of economic crops, such as soybean, aubergines, cassava, cotton, Brassicaceae plants, legumes, maize and other Poaceae, potatoes, sweet potatoes, and tomatoes, among other pot plants and vegetables [3,4,5]. Overall, approximately 200 host plants, belonging to 58 botanical families, including weeds, are recorded as associated with this species [3]. In a study documenting the occurrence and persistence of lepidopteran pests associated with soybean in the U.S., *S. eridania* was ranked as a secondary pest [6]. However, even though it is not considered a major pest on other host crops in the U.S., yield losses have been reported as a result of occasional outbreaks [7].

A key characteristic of *S. eridania* is its broad geographic distribution and invasive nature. Its range extends from South America to North America, including Canada [4,8,9]. In the United States, this species can migrate to northern regions during the warmer months, reaching as far as southern Ontario, Canada [9]. It overwinters in tropical and subtropical areas, with persistent populations occurring only in the southern states [10]. Although native to the American tropics, it has recently been reported as an invasive species in other parts of the world [11,12,13,14]. The invasive status of this species and its introduction to other continents were initially reported in 2016 and 2017, when it was observed defoliating cassava fields in West Africa (Nigeria and Bénin). In Central Africa (Cameroon and Gabon), it was later reported in tomato, amaranth, and maize [11,12]. In 2019, it was detected for the first time in a western state of India (Maharashtra), causing severe defoliation in soybean crops [12]. The international trade of fresh plant commodities between Asia and the Americas is considered the likely pathway for the westward invasion of this pest. This human-assisted pathway involves the transport of eggs and larvae with plants, fruits, cut flowers, or branches; pupae in soil or growing media; and adults as hitchhikers [10]. Weinberg et al. [9] refer to the species as a threat to global biosecurity, requiring effective phytosanitary measures to slow its global spread. In 2015, *S. eridania* was listed as an A1 quarantine pest recommended for regulation by the European and Mediterranean Plant Protection Organization [7]. In 2019, the European Food Safety Authority also classified it as an A1 quarantine pest/Annex II A [10].

In addition to its invasive behavior, larval capacity for economic damage, broad host range, and high adaptability, *S. eridania* has developed resistance to several insecticides [9,10,14,15,16]. Cases of resistance to genetically modified soybean and cotton expressing the Bt (*Bacillus thuringiensis*) toxin Cry1Ac have also been reported in Brazil [15,16], reducing the effectiveness of management tools already adopted in newly invaded regions, especially in Asia. Detection of this species in survey programs poses another challenge due to the high variability in larval appearance and the lack of strong contrasting patterns on the forewings of adults, making it difficult to distinguish *S. eridania* from native *Spodoptera* species in Africa and Asia [11]. These identification difficulties can lead to misdiagnosis or delayed management responses, facilitating the establishment and spread of the pest. Understanding the flight phenology of *S. eridania* is therefore crucial for optimizing monitoring efforts, timing control measures, and preventing further spread and resistance development.

One of the most important factors influencing *S. eridania* flight phenology and development is the weather conditions. Concerns about the increased suitability of new habitats for *S. eridania* and potential developmental changes that could complicate its management have been raised [14,17]. For instance, temperature plays a critical role in the species’ successful development, influencing voltinism, survival, and reproduction [17,18]. Models predicting the spatial expansion of *S. eridania* suggest that the presence of suitable pathways and habitats will likely facilitate its dispersal and establishment in new areas [13]. The expected geographic expansion is driven by the positive correlation between reproduction and temperature [18], indicating that regions currently just below the optimal temperature range may become suitable for reproduction in the near future. According to Zhang et al. [14], areas at higher elevations and latitudes are projected to become suitable habitats under future climate scenarios, with annual precipitation identified as the most influential factor affecting distribution. Furthermore, Sampaio et al. [17] projected a substantial increase in the number of generations per year as global temperatures rise, which could potentially lead to greater crop losses.

In this scenario, IPM/IRM programs should prioritize the proactive detection of early infestations of *S. eridania*. Understanding this species’ occurrence patterns and their association with local weather conditions is essential to improving the timing and precision of management interventions. Region-specific data must be developed to enhance forecasting capabilities and increase the success of management strategies. For example, analyses of environmental variables such as temperature and wind have been used to identify overwintering areas of related species like *S. exigua* [19], highlighting the value of incorporating climatic data into pest risk assessments [13,14]. Similar approaches could improve our ability to predict *S. eridania* movements, establishment potential, and outbreak risk under current and future climate scenarios. To our knowledge, no studies have documented the seasonal activity patterns or flight phenology of *S. eridania*, leaving a critical gap in understanding the temporal dynamics of this important pest. The objective of this study was to document the flight phenology of *S. eridania* in its native range of occurrence in an interbreeding region in subtropical America and identify the environmental variables influencing its population dynamics, based on year-long moth trapping data. Our main hypotheses were that (i) *S. eridania* exhibits a distinct seasonal peak in adult activity (abundance) in the region, (ii) there are inter-annual variations in both abundance and flight phenology of *S. eridania*, and (iii) environmental variables, such as temperature, precipitation, and wind patterns, can be used to predict peak abundance of this species.

## 2. Materials and Methods

### 2.1. Pheromone Trapping

Moths of *S. eridania* were monitored year-round from January 2018 to December 2023 using pheromone-baited delta traps. Four delta traps (Pherocon VI, Trécé Inc., Adair, OK, USA) were placed at a height of ~130 cm in four experimental fields (one trap per field, at least 600 m apart) at the West Florida Research and Education Center, University of Florida, located in Jay, in the northwest region of the Florida Panhandle (30.77622, −87.148833). Field areas ranged from 12.8 ha to 24.7 ha. A sex pheromone lure (Scentry Biological Inc., Oro Valley, AZ, USA) was added in each trap. Sticky cards within the traps were collected biweekly, and pheromone lures were replaced monthly. A total of 145 trapping dates were recorded over the six-year period. Pheromone-baited traps are designed to attract male moths; however, occasional captures of females are possible. We did not determine the sex of captured individuals, and no sex ratio was recorded for the population.

The surrounding landscape included a seasonal mosaic of agricultural and experimental areas, comprising corn (*Zea mays*), cotton (*Gossypium hirsutum*), soybean (*Glycine max*), and peanut (*Arachis hypogaea*) fields during the crop season (March through early December), as well as experimental plots of warm-season turfgrass, winter crops, and unmanaged fallow land. Corn, cotton, soybean, and peanut are usually planted in March and April and harvested between September and November. Weed species such as *Brassica carinata*, wild radish (*Raphanus raphanistrum*), common morning-glory (*Ipomoea purpurea*), and others were present at different times of the year. Among these, *I. purpurea* is a known host for *S. eridania* larvae [4], predominantly occurring throughout the crop season. There is no evidence supporting *B. carinata* or *R. raphanistrum* as larval host plants.

Weather data, including daily air temperature at 2 m, average and max wind speed at 10 m, and total precipitation at 2 m, were obtained from the Florida Automated Weather Network (FAWN) Jay Station located at the trapping site. The FAWN station uses a Campbell Scientific UT20 tower (Universal Manufacturing, Zelienople, PA, USA) equipped with a CR300/CR1000 data logger and sensors recording air temperature, relative humidity, rainfall, barometric pressure, solar radiation, wind speed, and wind direction. Measurements are taken at intervals of 5–15 s and reported every 15 min, with monthly summaries calculated for analysis. Data quality control involves range and step checks to ensure consistency within acceptable thresholds.

### 2.2. Statistical Analyses

All analyses were conducted in the software R version 4.3.1 [20]. Circular analysis was carried out using the package “circular” [21]. Generalized Linear Models (GLMs) or Generalized Linear Mixed-effect Models (GLMMs) were fit using the package “glmmTMB” [22]. Anova and emmeans functions from the package “car” [23] and “emmeans” [24], respectively, were used to perform Wald χ^2^ test statistics and to obtain marginal means. Model suitability was assessed with the “DHARMa” package [25].

To test the first hypothesis that there is a peak activity (significantly high abundance) of *S. eridania* in a specific period of the year, we ran a circular analysis and GLMM. We used circular statistical analysis to determine if moth abundance is concentrated at a specific time of the year. We calculated the mean vector (µ), mean vector length (r), median, and standard deviation, where the mean vector lengths (r) close to 1 indicate a peak abundance of moths concentrated around a single month or mean angle [26,27,28]. Watson’s goodness-of-fit test was used to assess whether the data were uniformly distributed. Since the pooled data did not follow a von Mises distribution, we used Watson’s goodness-of-fit test assuming a uniform distribution. We fit a GLMM to compare the total number of moths collected per trap per month and identify the months with a significant increase in moth captures. The number of moths was used as the response variable, month as the predictor variable, and year as a random effect to account for inter-annual variability. A negative binomial distribution was used to control overdispersion.

To test inter-annual variations in moth abundance and flight phenology (second hypothesis), we followed the same procedure previously described for the pooled data (all years combined) but now applied separately to each year. First, we used circular statistical analysis to determine whether moth abundance in each year was concentrated at a specific time of the year. Data from the six years did not follow a von Mises distribution; therefore, we applied Watson’s goodness-of-fit test under the assumption of a uniform distribution. Additionally, we used Watson’s two-sample test to assess whether the timing of peak moth abundance differed between years. Second, we fit GLMs of moth abundance across the year for each individual year.

To investigate the relationship between weather variables and *S. eridania* abundance (third hypothesis), we fitted GLMs with a negative binomial distribution to account for overdispersion. The response variable was the total number of moths per month per year. The predictor variables included average temperature (°C), total precipitation (mm), and maximum wind speed (km/h), all summarized per month per year. We compared a model containing only linear terms with an alternative model that included quadratic terms of each predictor variable to capture potential non-linear relationships between weather variables and *S. eridania* abundance. A likelihood ratio test indicated that the quadratic model provided a significantly better fit (χ^2^ = 17.96, *p* < 0.001) and had a lower corrected Akaike Information Criterion (AICc = 671.42) compared to the linear model (AICc = 682.0). Therefore, we selected the quadratic model for further analysis and interpretation.

## 3. Results

Over six years of *S. eridania* trapping, we collected a total of 2939 moths (Appendix A), with annual totals ranging from 237 to 731 moths. Circular analysis revealed a significant peak in *S. eridania* moth abundance at 258.05° (Figure 1a, Table 1), corresponding to September (Figure 1a). The mean number of *S. eridania* moths collected per trap varied significantly across months (χ^2^ = 158.05, *p* < 0.001; Figure 1b). September (11.2 ± 1.6 moths; mean ± SE) and October (11.3 ± 1.6 moths) had the highest trap counts, with at least 58% more moths on average than any other month. A slight increase in moth numbers was also observed in March (7.1 ± 1.2 moths, Figure 1).

There was a significant peak in moth abundance across all evaluated years (Figure 2, Table 1), with peak activity generally occurring between August and October. An exception was observed in 2019, when the peak occurred in April. Pairwise comparisons of moth distribution revealed significant differences between all years (Watson’s Two-Sample Test_range_ = 1.2–4.5, *p* < 0.001). Furthermore, the average number of moths collected varied significantly across months in each year (χ^2^_range_ = 75.1–163.7, *p* < 0.001; Figure 3). September and October were consistently the months with the highest numbers of moths collected; however, in some years, particularly 2019, 2022, and 2023, a significant increase in moth abundance was also observed from February to April.

Both the linear (χ^2^ = 19.89, *p* < 0.001) and quadratic terms for temperature (χ^2^ = 16.93, *p* < 0.001) significantly influenced the average number of moths collected per month per year (Figure 4, Table 2). A high number of *S. eridania* moths were captured during months with moderate average temperatures ranging from 15 °C to 26 °C (Figure 4). Neither the linear nor the quadratic terms for precipitation (linear: χ^2^ = 2.14, *p* = 0.14; quadratic: χ^2^ = 0.55, *p* = 0.45) or maximum wind speed (linear: χ^2^ = 0.92, *p* = 0.33; quadratic: χ^2^ = 0.99, *p* = 0.32) were significant.

## 4. Discussion

This study provides a long-term assessment of the seasonal dynamics and flight phenology of the southern armyworm, *S. eridania*, in the Florida Panhandle, based on six consecutive years of pheromone trapping data. Our results revealed the year-round presence of *S. eridania* moths and consistent activity patterns across years, with two distinct flight peaks and clear evidence that environmental variables significantly influence population fluctuations, corroborating our main hypotheses. These findings advance our understanding of the temporal dynamics of this pest and its implications for IPM and IRM in the region. By identifying periods of high adult activity, our results can inform the timing of monitoring programs and targeted control measures, such as insecticide applications or biological control releases, to coincide with population peaks. This temporal information can also help reduce unnecessary treatments during periods of low activity, thereby supporting resistance management by minimizing selection pressure on *S. eridania* populations.

Moths of *S. eridania* exhibited year-round flight activity in the Florida Panhandle, with seasonal fluctuations in abundance. This continuous presence suggests that populations are sustained locally throughout the year, likely supported by the pest’s ability to feed on a wide range of cultivated and wild host plants [4,29]. Such ecological flexibility enables *S. eridania* to exploit alternative hosts during periods when major crops are unavailable, contributing to its persistence across seasons. This seasonal flight pattern is consistent with the biology of other lepidopteran pests in the region, including the fall armyworm (*Spodoptera frugiperda*) and corn earworm (*Helicoverpa zea*), which are known to overwinter in southern Florida [1,30,31,32,33,34] and migrate northward as environmental conditions become favorable [2,31,34,35,36,37,38]. The flight behavior among these species highlights the importance of regional monitoring programs that consider multiple pest species with similar ecological strategies, including new areas where this species is currently an invasive pest.

Two prominent peaks in *S. eridania* activity were consistently detected throughout the study period. The first less intense peak occurs in early spring, around March, coinciding with the transition between winter and spring cropping systems in southern Florida, a period when host plant availability may decline. This timing suggests that moths may disperse northward from southern regions as host crops become scarce in the region. Although *S. eridania* has been reported as a migratory species in locations such as Bermuda [39], its occurrence suggests a strong potential for regional migratory behavior within the southern U.S., similarly to the movement patterns observed in *S. frugiperda* [30,40,41]. Like *S. frugiperda*, which does not undergo diapause and overwinters in southern Florida and southern Texas, *S. eridania* may also take advantage of favorable climatic conditions to disperse northward as temperatures rise in early spring. In *S. frugiperda*, these overwintering populations begin migrating northward with warm temperatures [31,34,38], typically reaching the Florida Panhandle, an important interbreeding zone for this species [31,34]. Therefore, the early-season peak in *S. eridania* activity observed in the FL Panhandle likely reflects regional dispersal within Florida, potentially driven by the end of the crop season of host vegetables cultivated in southern Florida and the availability of more favorable conditions and host plants in northern areas. However, we acknowledge that our study did not directly investigate flight behavior or dispersal directionality, and further research incorporating spatial trap distribution, wind patterns, and landscape connectivity would be necessary to confirm these hypotheses. The second more pronounced peak of flight activity occurs between September and October, toward the end of the crop season in the Florida Panhandle, likely reflecting population buildup in locally grown summer crops, particularly in soybean and peanut. Soybean has been shown to be especially suitable for *S. eridania* development and oviposition, supporting higher survival and nutritional performance [42], which may contribute to the higher moth abundance observed during this period.

This bimodal flight activity has also been observed in other *Spodoptera* species [19], suggesting that it may reflect favorable environmental windows for development and reproduction [9,17,18]. For instance, a flight phenology study of *S. exigua* in California reported similar bimodal peaks aligned with specific months of the year [19]. In our study, both *S. eridania* flight peaks occurred under moderate temperatures (23–27 °C), consistent with the optimal thermal conditions reported for this subtropical species [9,17,18,43]. While *S. eridania* can complete development between 15 and 32 °C [10,17], peak oviposition and reproductive performance typically occur between 20 and 26 °C [18], matching the range in which we observed the highest moth abundance. Temperatures above 28 °C tend to reduce adult longevity and accelerate pre-oviposition, whereas egg hatch is inhibited above 34 °C [17,18]. Based on these results, the pest’s wide host range and thermal adaptability further support its persistence across seasons in the southeastern U.S.

Since crop availability and weather conditions are directly linked to the flight activity of *S. eridania* moths, they likely explain the observed interannual variation in peak abundance. Despite consistent seasonal trends, substantial year-to-year differences in peak moth numbers were evident. This suggests that, beyond crop phenology and host availability, weather factors play a significant role in shaping population dynamics. Specifically, a negative quadratic relationship between moth abundance and temperature was detected, indicating that population levels peak within a moderate temperature range (approximately 15–26 °C; Figure 4). This finding aligns with laboratory and field studies that identify this temperature window as optimal for both larval and adult development [43], as discussed previously. Additionally, population carryover effects from previous seasons, such as residual local populations or the influx of migrants from other regions, could contribute to annual differences in abundance and phenology. Although the number of generations per year and the relative contributions of local reproduction versus migration have not been quantified for this species in our region, the subtropical climate likely allows for multiple overlapping generations. Future studies incorporating field life tables, population genetic analyses, and modeling approaches would be valuable to disentangle the relative importance of these drivers in shaping the population dynamics of *S. eridania* in the region.

Our study provides important insights into the flight phenology and environmental drivers of *S. eridania* populations, revealing predictable periods of high pest pressure influenced by weather conditions, particularly temperature. This knowledge enables more precise timing of scouting and control measures, thereby improving the effectiveness of regional IPM and IRM strategies. By documenting the seasonal dynamics and environmental conditions that favor outbreaks, our findings support the development of forecasting models designed to mitigate economic damage [9,14]. Beyond the regional context, the implications of this research are global as *S. eridania* has emerged as an invasive pest, having spread to parts of West and Central Africa and South Asia [11,12]. Its recent reports in new regions, including Europe, where it is now a quarantine pest [10], highlight the urgency of effective management. Risk assessments indicate that *S. eridania* could expand further into eastern Oceania and western Europe [14]. Our long-term dataset from the species’ native range thus provides critical baseline information essential for predicting its spread, enhancing global IPM and IRM programs to safeguard host crops of this pest.

## Figures and Tables

**Figure 1 insects-16-00779-f001:**
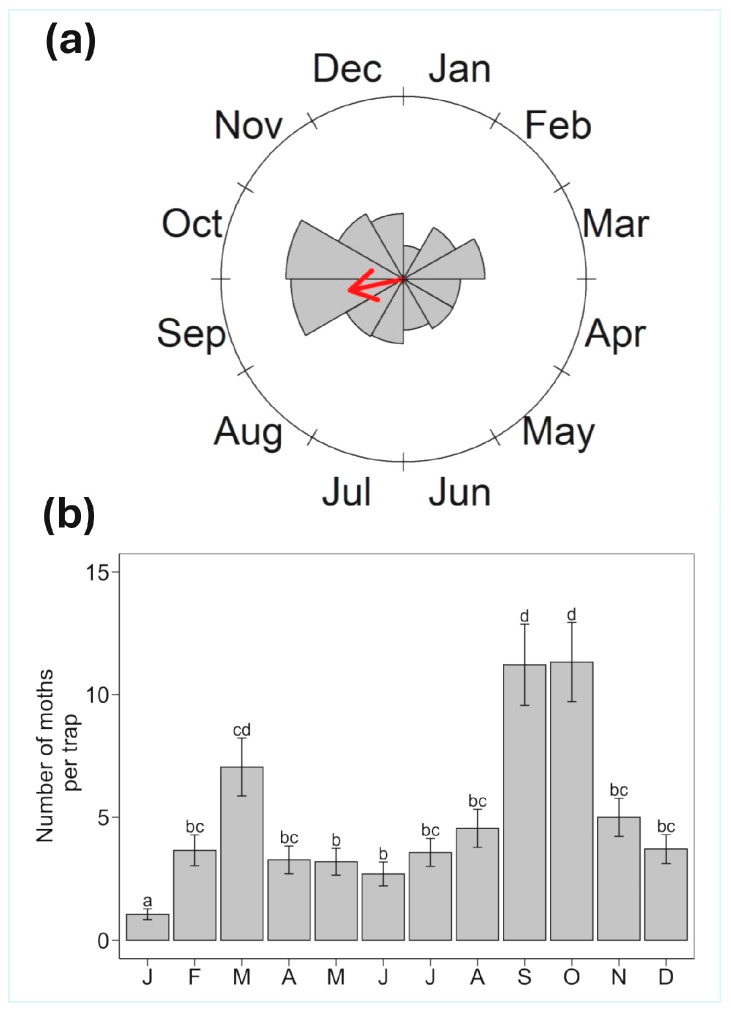
Flight phenology of *S. eridania* moths in the Florida Panhandle based on six years of data. (**a**) Significant peak in *S. eridania* moth abundance throughout the year based on circular analysis. The red arrow represents the mean vector (r = 0.30), indicating the mean angle of peak abundance in September. Arrow length reflects the concentration of moth captures around the mean angle, with longer arrows indicating greater concentration. (**b**) Monthly variation in *S. eridania* moth abundance. Different letters indicate significant differences between months based on Tukey’s HSD test. Circular statistical results are detailed in Table 1.

**Figure 2 insects-16-00779-f002:**
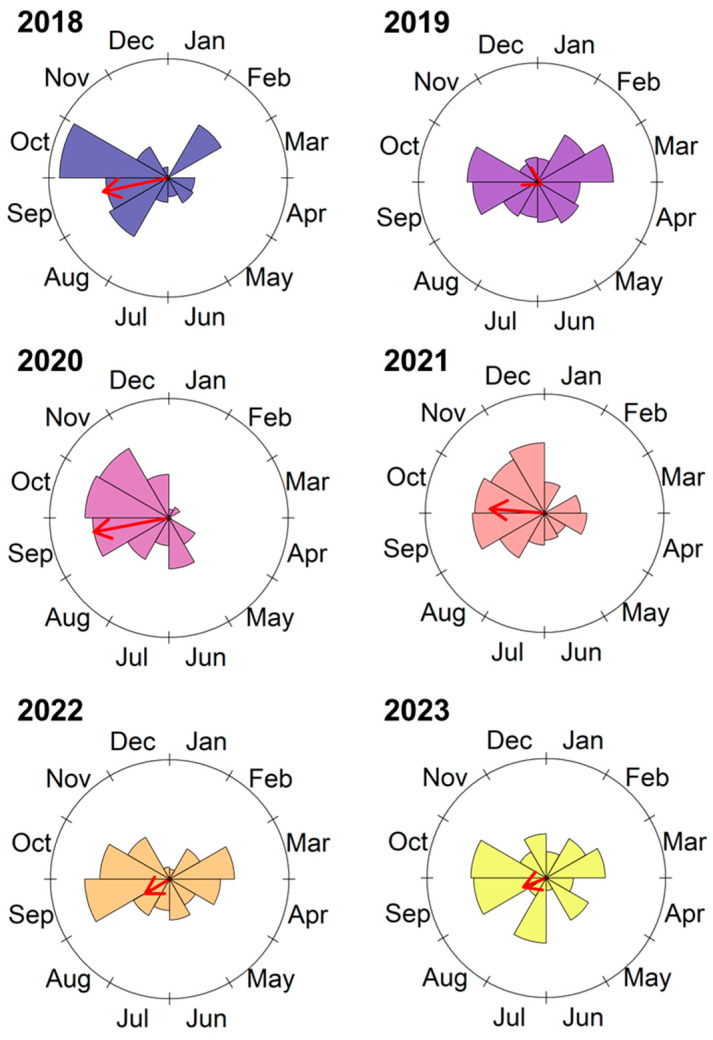
Flight phenology of *S. eridania* moths in the Florida Panhandle for each evaluated year. Red arrows represent the mean vector, indicating the mean angle of peak moth abundance. This represents the average timing of abundance across the year, rather than the month with the highest raw count. Arrow length reflects the concentration (r) of captures around the mean angle, with longer arrows denoting greater concentration. Circular statistical results are presented in Table 1.

**Figure 3 insects-16-00779-f003:**
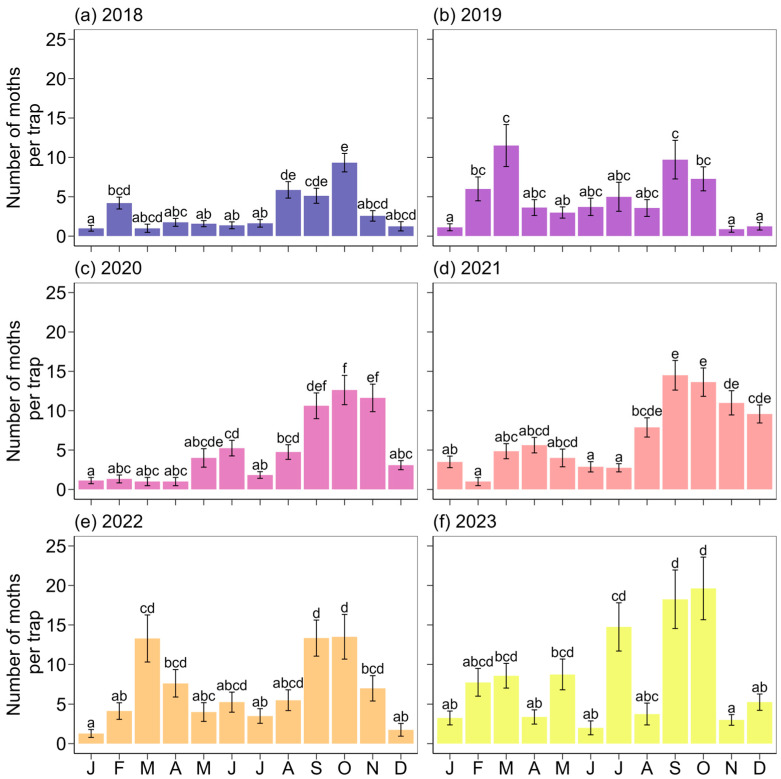
Monthly variation in *S. eridania* moth abundance in the Florida Panhandle for each evaluated year: (**a**) 2018, (**b**) 2019, (**c**) 2020, (**d**) 2021, (**e**) 2022, and (**f**) 2023. Different letters in each panel indicate significant differences between months based on Tukey’s HSD test.

**Figure 4 insects-16-00779-f004:**
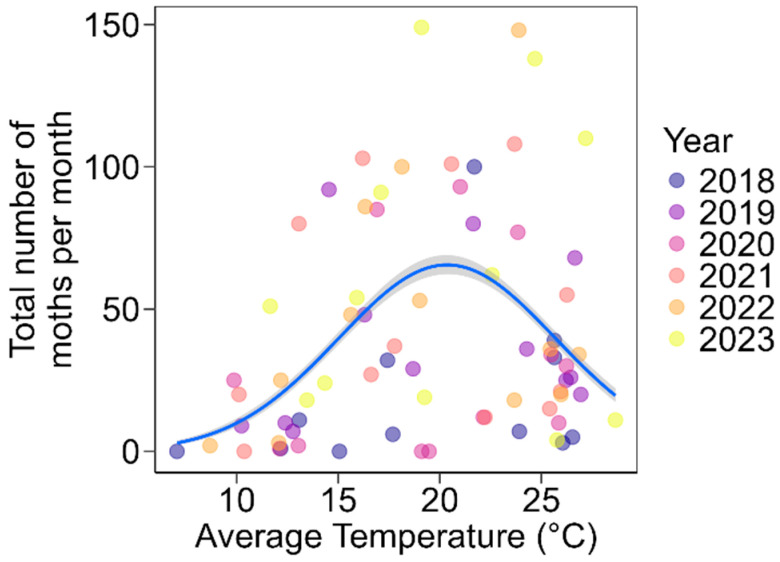
Significant non-linear (quadratic) relationship between the total number of *S. eridania* moths and the average monthly temperature per year. The solid blue line represents the best-fit curve from the non-linear regression, with standard error.

**Table 1 insects-16-00779-t001:** Circular statistical analysis of *S. eridania* moth abundance across years and for each year of trapping. Watson U^2^ test was performed with uniform distribution.

Year	Mean Vector (Degrees)	Mean Month	Vector Length (rho)	Watson U^2^ Test	*p*-Value
All years	258.05	Sep	0.30	7.47	<0.01
2018	258.01	Sep	0.55	5.21	<0.01
2019	117.32	Apr	0.04	1.57	<0.01
2020	259.51	Sep	0.64	8.84	<0.01
2021	274.56	Oct	0.46	7.19	<0.01
2022	238.9	Aug	0.24	4.21	<0.01
2023	247.97	Sep	0.21	3.37	<0.01

**Table 2 insects-16-00779-t002:** GLM results of the linear and quadratic effects of temperature, precipitation, and max wind speed on the total number of *S. eridania* captured. Bold *p*-values represent significant results.

Predictor	Estimate	SE	Degrees of Freedom	χ^2^	*p*-Value
Temperature	0.764	0.171	1	19.8	**<0.001**
Temperature^2^	−0.018	0.004	1	16.9	**<0.001**
Precipitation	−0.004	0.003	1	2.1	0.144
Precipitation^2^	0.000	0.000	1	0.5	0.457
Max wind speed	−0.082	0.086	1	0.9	0.338
Max wind speed^2^	0.001	0.001	1	0.9	0.320

## Data Availability

The original contributions presented in this study are included in the Appendix A. Further inquiries can be directed to the corresponding author.

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
