# Peer review of "Flight Phenology of *Spodoptera eridania* (Stoll, 1781) (Lepidoptera: Noctuidae) in Its Native Range: A Baseline for Managing an Emerging Invasive Pest"

_insects, 2025, doi:10.3390/insects16080779_

Round 1

Reviewer 1 Report

Comments and Suggestions for Authors

Dear authors,

in the appendix you will find some, I hope, useful comments and guidelines.

I find the results of your research very useful and highly recommend your article for publication.

Author Response

Reviewer: #1

In my opinion, the manuscript is very well written. I have no reservations except for minor formal corrections needed in the "References" section and one note, which can be found at the end of these comments.

Response #4: Thank you for your positive words and suggestions, which we have addressed below.

Dear authors, make sure to edit references in accordance with the journal's author guidelines. See, for example, References of the latest articles published in the Insects for inspiration. Below are a couple of issues that need editing:

Response #5: We have now edited our references according to the journal’s guidelines. According to them, the words of the title of articles are written with capital letters, except the epithet of species. Since we are using Zotero software to edit the citations, the program keeps changing the epithet to capital letter. We are aware of this issue and will make sure the final version for publication has the correct format for species name, if the paper is accepted for publication.

Where possible, the titles of cited articles should be abbreviated.

Response #6: We now abbreviated the journals’ title.

Scientific names of organisms in the title of articles cited should be written: genus name with a capital letter, species name with a lower case letter, e.g., Spodoptera eridania instead of Spodoptera Eridania.

Response #7: Done. Also see our Response #5.

And the titles of the cited articles should be without capitals, e.g., „Spodoptera Eridania: Current and Emerging Crop Threats from Another Invasive, Pesticide-Resistant Moth. entomologia 2022, 42, 701–712, doi:10.1127/entomologia/2022/1397” by Weinberg, J.; Ota, N.; Goergen, G.; Fagbohoun, J.R.; Tepa-Yotto, G.T.; Kriticos, D.J. should be written as “Spodoptera eridania: Current and emerging crop threats from another invasive, pesticide-resistant moth. Entomologia 2022, 42, 701–712, doi:10.1127/entomologia/2022/1397”

Response #8: Done

Line 342: the ending of the citation: Pogue, M.G. A World Revision of the Genus Spodoptera Guenée (Lepidoptera: Noctuidae); American Entomological Society Philadelphia, 2002; Vol. 43;. ?

Response #9: Done

Line 346: the article title should be with a capital “Entomologia”

Response #10: Done

Line 366 and 367: “Machado, E.P.; Dos S Rodrigues Junior, G.L.; Somavilla, J.C.; Führ, F.M.; Zago, S.L.; Marques, L.H.; Santos, A.C.; Nowatzki, T.; Dahmer, M.L.; Omoto, C.; et al. Survival and Development of…” – “Dos” should be written in a lowercase letter “dos” and “et al.” should be replaced with the name of the last author Bernardi, O.

Response #11: We followed the journal’s guideline and kept the et al.: “For documents co-authored by a large number of persons (more than 10 authors), you can cite the first ten authors, then add a semicolon and add ‘et al.’ at the end: Author 1; Author 2; Author 3; Author 4; Author 5; Author 6; Author 7; Author 8; Author 9; Author 10; et al. ”

Line 336 and 392 (citations 5 and 26): maybe there is a semicolon lacking in front of the ISBN in the citation no. 26 when comparing with the citation no. 5

Response #12: According to journal’s guidelines, there is no semicolon after ISBN number.

Line 404: the ending of the citation: Luginbill, P. The Fall Army Worm; US Department of Agriculture, 1928; ?

Response #13: Done.

Line 406: “flen” – Florida Entomologist – use appropriate abbreviation of the journal, please

Response #14: We have now changed to Fla. Entomol.

In addition, the taxonomical data should be corrected, since the official year of description is 1781, not 1782 (line 44): “Spodoptera eridania (Stoll, 1781) is the current valid name (ITIS, 2019) of a highly polyphagous herbivorous moth (Lepidoptera: Noctuidae) native to the American tropics. This species has many synonyms (Todd and Poole, 1980) including the presently invalid authority name ‘Cramer’ (i.e. Spodoptera eridania (Cramer, 1784)), which is the one listed in Annex IAI of Council Directive 2000/ 29/EC.”

Reference: EFSA Panel on Plant Health (PLH), Bragard, C., Dehnen‐Schmutz, K., Di Serio, F., Gonthier, P., Jacques, M. A., ... & MacLeod, A. (2020). Pest categorisation of Spodoptera eridania. EFSA Journal, 18(1), e05932. https://efsa.onlinelibrary.wiley.com/doi/pdfdirect/10.2903/j.efsa.2020.5932

Response #15: Done.

In the end, I have a question. You used pheromone traps to attract male moths, as written in (and only) the “Simple Summary” section. Then, in the section “Results” you are referring to 2,939 moths that you collected during the research. And my question is: were all the moths trapped on sticky cards males? Or was there any female caught by chance? Or were there more females caught because they might have been attracted by males glued on these cards? What is the sex ratio of this species if it is known? I mean, you only collected male moths, but throughout the text it seems as if entire populations of moths consisting of males and females were being evaluated. I suggest, therefore, that you mention this fact in the Results and Discussion, or at least explain it in more detail in the Methodology that you investigated the population of males, but the results found are applicable to the entire moth population, i.e., also to females. Or something along those lines.

Response #16: We agree. Although pheromone traps are designed to attract males, many females are also captured. We acknowledge that our dataset likely includes some females; however, we did not identify the sex of the moths, nor do we have information on the population sex ratio. We have revised the manuscript to include this clarification in detail in the simple summary and methods (L. 16, 135-137).

Reviewer 2 Report

Comments and Suggestions for Authors

This paper just shown the population dynamics of southern armyworm (SAM), and don't have any further analysis.

(i) The histograms have shown very clear patterns, I don't think circular analysis shown more interesting result, so just one kind of figure is enough.

(ii) Readers would have more interesting about the drivers of the seasonal and annual population variations: Why some year have more moths, why some year moths appear earlier? This paper have tried this, but I think still need more improvement. Firstly, more detail of the weather data should be provided in method; Secondly, more variables should be considered, like the host plants, the number in previous season/moth. Thirdly, how many generations in this regions? The population of each generation is migrated from other place or local reproductions? The factor driver of each generation might be different. Anyway, I think more analysis should be included.  

Author Response

Reviewer: #2

This paper just shown the population dynamics of southern armyworm (SAM), and don't have any further analysis.

Response #17: Thank you for your comments and suggestions, which we have addressed below.

(i) The histograms have shown very clear patterns, I don't think circular analysis shown more interesting result, so just one kind of figure is enough.

Response #18: We agree that the histograms show clear patterns. However, we believe the circular analysis provides additional value by quantitatively summarizing the seasonality (e.g., mean vector, concentration, and tests of uniformity), and comparing peak of activity across years. Therefore, we cordially ask to keep the circular analysis and figures in the manuscript.

(ii) Readers would have more interesting about the drivers of the seasonal and annual population variations: Why some year have more moths, why some year moths appear earlier? This paper have tried this, but I think still need more improvement. Firstly, more detail of the weather data should be provided in method; Secondly, more variables should be considered, like the host plants, the number in previous season/moth. Thirdly, how many generations in this regions? The population of each generation is migrated from other place or local reproductions? The factor driver of each generation might be different. Anyway, I think more analysis should be included. 

Response #19: We have now provided more detailed information about the weather data in the Methods section (L. 148–156). Conducting the additional analyses suggested would require field life table studies and mathematical modeling, which were beyond the scope of the present study. We agree that factors such as the number of generations and moth abundance in previous seasons can influence population dynamics; however, additional data on key life history parameters (e.g., sex ratio, egg production, and larval and adult survival rates) would be necessary to support such conclusions. We have added more discussion on these limitations and potential drivers of seasonal variation to acknowledge these important points (L. 326–334).

Reviewer 3 Report

Comments and Suggestions for Authors

Please find the attached report.

Author Response

Reviewer: #3

The manuscript provides valuable data on the S. eridania flight phenology and seasonal dynamics using pheromone trapping to evaluate population trends and environmental drivers. A notable strength of this study is its six-year duration, which enables the evaluation of interannual variability in moth population dynamics.

Response #20: Thank you for your positive words and suggestions, which we have addressed below.

However, several areas require some revisions in the manuscript sections. Below, I outline the specific areas of concern and provide recommendations for improvement.

2 - Spodoptera eridania -Please add a descriptor, family, and order of Spodoptera eridania

Response #21: Done

2 - Suggest revising the title to focus on “adult phenology” instead of “flight patterns” to better match the study’s actual data on seasonal trap captures.

Response #22: We changed the title to “Flight phenology of Spodoptera eridania (Stoll, 1781) (Lepidoptera: Noctuidae) in its native range: A baseline for managing an emerging invasive pest”.

23 - Please add a descriptor, family, and order of Spodoptera eridania

Response #23: Done

33-35 - I recommend revising this statement. The authors could rephrase it to say that the study provides foundational information that could support future monitoring tools for IPM programs.

Response #24: Rephrased as suggested.

75-84 - This paragraph is not directly related to the research topic, as it focuses more on the insect’s pesticide resistance and identification traits, whereas the study should focus on the main research topic (adult flight phenology). Therefore, I suggest the authors review this paragraph or remove it if it is not directly relevant to the study’s objectives.

Response #25: Our intention in the introduction and in this paragraph was to provide a broader context about S. eridania as an invasive, polyphagous, and migratory pest with documented insecticide resistance. We believe understanding the adult flight phenology is critical for anticipating and managing these issues effectively. We have now added a final sentence in the paragraph better linking the relevance of studying the flight phenology of S. eridania and the resistance to insecticides (L. 87-89).

113-114 - The 1st objective should be more specific. Although S. eridania exhibits a distinct seasonal peak in adult abundance in the region, the study did not include sampling of eggs or larvae. The statement about the timing of egg and larval sampling in host crops is not directly supported by the data presented. I recommend revising the objective to reflect the sampling scope to study the seasonal peak in adult abundance.

Response #26: We removed the second part of the hypothesis which mentions the egg and larva number.

I encourage the authors to expand the introduction by including information on the seasonal activity patterns of S. eridania from other parts of Florida, the U.S., or even globally. This additional context would help readers better understand why it’s important to study the seasonal dynamics of this species in the current region and how these results relate to what’s already known from other areas.

Response #27: To our knowledge, there are no published studies evaluating the flight phenology of S. eridania in Florida, the U.S., or other parts of the world. This study represents the first effort to describe the seasonal activity patterns of this species, and we have highlighted this gap in the Introduction to emphasize the novelty and importance of our work (L.114-116).

122-125 - This sentence is quite long and could be broken up for clarity. It’s also not clear whether the four pheromone lures were different types or simply replicates of the same lure. The methodology does not indicate how many traps were deployed across the experimental fields. Were the four pheromone lures each deployed in a single trap, or were multiple replicate traps used? Add the area of the specific experimental field, and the number of traps per field (or per hectare), and the spatial arrangement (grid, random, transect, edge vs. center). Also, specifying the spacing between traps would improve the description of the methodology.

Response #28: Lures from the same company was used during the entire experiment. Four delta traps were used in four fields, one trap per field. We have updated this information in the manuscript. We also added more information about the area of each field (L. 127-137).

129-134 - I recommend adding a table that lists the crops grown in the experimental fields in the Florida Panhandle, including their scientific names. It would also be helpful to include information on the weed species present and the phenology (growth stages) of the crops during the study period. The authors note the presence of Brassica carinata, wild radish, and other weeds. It might be useful to indicate whether these are known hosts for S. eridania larvae or nectar sources for adults, as this can influence local moth populations and trap captures.

Response #29: The study was conducted over six consecutive years and encompassed the full annual cycle, including all phenological stages of crops and weeds. Unfortunately, we do not have detailed records of the specific crops planted in each field or their phenological stages throughout the study period when traps were evaluated. However, we have now added more information into the manuscript about the host plants and surrounding landscape (L.138-147).

137-176 - The manuscript provides insufficient justification for the analytical choices and data transformations applied. I recommend that the authors clearly explain the rationale behind these decisions to make the analysis easier to follow and more transparent.

Response #30: We have rephrased some parts of the data analysis to clearly explain the rationale for each analytical choice (L. 165-166). No transformation was done in our data.

185 - The circular plot (Fig. 1a) shows a red mean vector, but does not include the r statistic (mean vector length), which is essential for interpreting concentration strength. This should be added to the figure or at least annotated.

Response #31: We have now added the rho value in the legend of the figure.

206-214 - Figure 2 indicates that the circular plot suggests a peak around September (as indicated by the arrow); however, Figure 3 suggests the bar plot shows October as slightly higher. The figure legend or discussion should explain this, perhaps noting that the circular analysis weights the distribution differently (i.e., concentrates timing) versus raw counts.

Response #32: We have now added more information about the meaning of the direction of the arrow in the legend of the figure 2.

224-226 - Figure 4 effectively illustrates the significant quadratic relationship between S. eridania abundance and average temperature. The manuscript should include a corresponding table summarizing the model results for all abiotic predictors, including precipitation and wind speed. Although the text reports the χ² statistics and p-values for these additional factors, a formal table presenting: model coefficients (estimates ± SE), test statistics (χ² -values), degrees of freedom, and exact p-values.

Response #33: We added a table with model results (Table 2).

233-234 - While the manuscript concludes that these findings advance our understanding of the temporal dynamics of S. eridania and have implications for IPM and IRM in the region, this connection is not demonstrated by the data presented. I recommend adding a short conceptual discussion on how the documented phenology could be used to optimize monitoring or control timing, to more clearly establish the relevance to IPM and IRM.

Response #34: We now added more information in the discussion of how our results can help optimize monitoring or control timing (L. 260-265).

255-263 - The manuscript proposes that the early-season peak in S. eridania activity may reflect regional migratory behavior similar to S. frugiperda, but the study does not directly investigate or present data on actual flight behavior or dispersal directionality. I recommend providing additional discussion on these limitations to avoid overextending the conclusions. For example, analysis of the spatial distribution of trap captures across the study region, GPS coordinates of individual trap sites, or prevailing wind patterns and landscape corridors that might support hypotheses of immigration.

Response #35: We have now acknowledged this limitation of our study and conclusions in the discussion (L. 295-298).

265-268 - I recommend including a table in the methodology that summarizes the major crops (e.g., soybean, peanut, ….) and unmanaged areas present in the study landscape, with their seasonal presence or phenology that may support year-round populations.

Response #36: We have now added more information about the landscape and crop phenology in the methods (L. 138-147). Also see our response #29.